# Merging the Versatile Functionalities of Boronic Acid with Peptides

**DOI:** 10.3390/ijms222312958

**Published:** 2021-11-30

**Authors:** Yahong Tan, Junjie Wu, Lulu Song, Mengmeng Zhang, Christopher John Hipolito, Changsheng Wu, Siyuan Wang, Youming Zhang, Yizhen Yin

**Affiliations:** 1State Key Laboratory of Microbial Technology, Institute of Microbial Technology, Shandong University, Qingdao 266237, China; tanyahong@sdu.edu.cn (Y.T.); wujunjie@mail.sdu.edu.cn (J.W.); songlulu@mail.sdu.edu.cn (L.S.); zhangmengmeng@mail.sdu.edu.cn (M.Z.); wuchangsheng@sdu.edu.cn (C.W.); zhangyouming@sdu.edu.cn (Y.Z.); 2Screening & Compound Profiling, Quantitative Biosciences, Merck & Co., Inc., Kenilworth, NJ 07033, USA; chris.john.hipolito@gmail.com; 3Department of Medicinal Chemistry, College of Pharmacy, Shenzhen Technology University, Shenzhen 518118, China

**Keywords:** boronic acid, peptide, inhibitors, molecular recognition, synthesis, modifications

## Abstract

Peptides inherently feature the favorable properties of being easily synthesized, water-soluble, biocompatible, and typically non-toxic. Thus, boronic acid has been widely integrated with peptides with the goal of discovering peptide ligands with novel biological activities, and this effort has led to broad applications. Taking the integration between boronic acid and peptide as a starting point, we provide an overview of the latest research advances and highlight the versatile and robust functionalities of boronic acid. In this review, we summarize the diverse applications of peptide boronic acids in medicinal chemistry and chemical biology, including the identification of covalent reversible enzyme inhibitors, recognition, and detection of glycans on proteins or cancer cell surface, delivery of siRNAs, development of pH responsive devices, and recognition of RNA or bacterial surfaces. Additionally, we discuss boronic acid-mediated peptide cyclization and peptide modifications, as well as the facile chemical synthesis of peptide boronic acids, which paved the way for developing a growing number of peptide boronic acids.

## 1. Introduction

Boron, an element barely observed in a living body, has exhibited considerable potentials in drug discovery, chemical biology, organic chemistry, and material sciences [1,2,3,4,5]. In particular, boron has continually drawn much attention from medicinal chemists for many decades, although it is also toxic in excessive amounts [1,2,6]. The utility of boron arises from the empty 2*p* orbital, which allows boron to coordinate heteroatoms such as oxygen and nitrogen [1,7]. Among several boron-based functional groups, boronic acid represents a versatile and robust functional group. In addition to their synthetic utility, boronic acid compounds can also form reversible covalent interactions with amino acid side chains in proteins or diol group in sugars [1,7]. Such properties allow chemists to employ boronic acids as reversible covalent warheads for medicinal and biological applications [8]. Furthermore, boronic acid can also be considered as a bioisostere for carboxylic acids in drug design [9,10].

Peptides inherently feature favorable characteristics, such as the relative ease of synthesis (e.g., solid-phase peptide synthesis), acceptable water solubility, biocompatibility, and little to no toxicity [11,12,13,14]. Therefore, boronic acids can be incorporated into peptides for the discovery of peptide ligands with novel biological activities by merging both merits of boronic acids and peptides. For instance, the approval of peptide boronic acid medicines, such as bortezomib and ixazomib, as multiple myeloma drugs, underscores the notable potential of peptide boronic acids in drug discovery [6,15,16] (Figure 1). Taking advantage of its covalent reversible binding with amino acids or sugars, peptides boronic acids can be widely utilized as covalent reversible enzyme inhibitors, ligands, or carriers of glycans and RNA, etc. [1,2,3,4,5]. Additionally, the integration of boronic acids provides a straightforward strategy for peptide modifications or conjugation [17,18]. Moreover, peptide scaffolds bearing boronic acids have also been attractive delivery vehicles of ^10^B (boron-10) agents for boron neutron capture therapy (BNCT), as well as efficient precursors of direct radioiodination for imaging applications [19].

Given the considerable potential of boronic acids, it is vital to review the recent advances of boronic acid as a versatile functional group. Reviews in the past decade have focused on the wide applications of boronic acid-based small molecule drugs [1,2,4,6,20,21], molecular recognition and sensing [22,23,24,25,26,27,28,29], polymers [3], protein modifications [17], bioconjugation [18], and dynamic click chemistry [30]. Herein, we prioritize the integration between boronic acid and peptides, an important but largely overlooked perspective.

Dramatic advances have been witnessed in this field since 2011, and thus, we provide an overview of the versatile and robust functionalities of boronic acid by its integration with peptides. In this review, we summarize various applications of peptide boronic acids in medicinal chemistry and chemical biology, including the identification of covalent reversible enzyme inhibitors, recognition, and detection of glycans on proteins or the cancer cell surface, delivery of siRNAs, development of pH-responsive devices, and recognition of RNA or bacterial surfaces. Additionally, we discuss boronic acid-mediated peptide cyclization, dipeptide synthesis, and peptide modifications, as well as the facile chemical synthesis of peptide boronic acids, which have paved the way for developing a growing number of peptide boronic acids.

## 2. Peptide Boronic Acids as Enzyme Inhibitors

Peptide boronic acids as covalent reversible enzyme inhibitors have gained increasing attention from medicinal chemists. It is believed that such a covalent reversible binding mode would confer the inhibitor with slower dissociation from the target protein, and therefore a longer “residence time”. Sauvage’s group solved the crystal structures of a penicillin-binding protein (PBP, the DD-peptidase from *Actinomadura sp. R39*) with five peptide boronic acids (**1**–**5**, Figure 2), and found that all of the five compounds could form a tricovalent adduct with three key residues (hydroxyl groups of Ser49 and Ser298, as well as the terminal amine group of Lys410) residing in the catalytic site of PBP [31]. Interestingly, monocovalent transition state analogue structures were also observed for two of the five peptide boronic acids, suggesting that the compounds might first associate with the active site serine and then rotate and move close to finally form the tricovalent binding mode for PBP inhibition [31,32]. Owing to the covalent reversible binding to amino acids, a great number of peptide boronic acids have been developed as inhibitors against diverse enzymes, including serine proteases, threonine proteases, aspartyl proteases and arginases, in the past ten years.

### 2.1. Serine Protease Inhibitors

Peptide boronic acids have been demonstrated to be very potent inhibitors against serine proteases, which are a class of enzymes with serine as the nucleophilic amino acid at the active site (Figure 3). hClpXP is an ATP-dependent serine protease in the mitochondrial matrix. Santos’s group reported the synthesis and development of a range of N-terminal peptide boronic acids as potential protease inhibitors. WLS6a (**6**) was identified as the most effective inhibitor of the series with the IC_50_ value of 29 µM (Table 1) [33]. Furthermore, WLS6a demonstrated selective inhibition against hClpXP, providing a useful tool for investigating the physiological roles of hClpXP in the maintenance of mitochondrial integrity [33]. Another peptidyl boronic acid, MG262 (**7**), was also found to inhibit the ATP-dependent peptidase activity of hClpXP. In contrast to WLS6a, the boronic acid moiety of MG262 is located at the C-terminus (Table 1) [34].

Fibroblast activation protein (FAP), a serine protease highly abundant on activated stromal fibroblasts epithelial carcinomas, has been widely accepted as a potential new therapeutic target due to its critical role in tumor invasion and metastasis [35]. However, it is challenging to achieve the highly specific inhibitors for FAP because of the simultaneously competitive inhibition against the other serine proteases, such as the dipeptidyl peptidases (DPPs) and prolyl oligopeptidase (PREP). For instance, Val-boroPro (Talabostat, **8**), an orally available DPP4 inhibitor (IC_50_ < 4 nM; K_i_ = 0.18 nM) and the first clinical FAP inhibitor (IC_50_ = 560 nM) [36,37], has also recently been found to inhibit two post-proline-cleaving serine proteases DPP8/9 (Table 1) [38]. By inhibiting DPP8/9, the inflammasome sensor protein Nlrp could be activated followed by the activation of pro-protein form of caspase-1. The activated form pro-caspase-1 could subsequently activate gasdermin D to induce pyroptosis in monocytes and macrophages, leading to powerful anti-tumor immune responses in syngeneic cancer models [39,40]. To obtain highly specific FAP inhibitors, Poplawski et al. identified N-(pyridine-4-carbonyl)-D-Ala-boroPro (ARI-3099, **9**) as the first potent FAP inhibitor with high selectivity over the DPPs by 100-fold and PREP by 360-fold (Table 1) [41,42]. Meanwhile, a similarly potent and selective PREP inhibitor, N-(pyridine-3-carbonyl)-Val-boroPro (ARI-3531, **10**), was also obtained with 77000-fold selectivity over both the DPPs and FAP (Table 1).

The flavivirus proteases, which belong to the trypsin-like serine protease family, are attractive targets for developing antiviral drugs. Nitsche et al. introduced a covalent reversible boronic acid building block into the C-terminus moiety within peptidic inhibitors of flaviviral proteases. The resulting compounds, such as molecule **11**, have exhibited a considerable increase of affinity by a factor of 1000 as compared to their respective amide congeners (Table 1) [43]. Several compounds were further demonstrated to be effective in cell-culture models of dengue virus (DENV) and West Nile virus (WNV) replication with low cytotoxicity [43]. Although the boronic acid moiety tends to impair the cellular uptake of the compounds, which might lead to relatively weak activity of the compounds in cellular assays, it provides an alternative strategy for developing novel antiflaviviral drugs through the introduction of boronic acids.

**Table 1 ijms-22-12958-t001:** Representative peptide boronic acids have been used as serine protease inhibitors since 2011.

Compound	Structure	Target	Pathology
**6**(WLS6a)	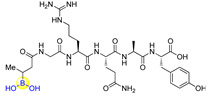	hClpXP	IC_50_ of hClpXP = 29 μM [33]; proteolytic cleavage of casein mediated by hClpXP was clearly inhibited as demonstrated by in vitro study [33].
**7**(MG262)	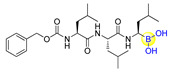	hClpXP	IC_50_ of hClpXP ≈ 40 μM [34].
**8**(Talabostat, Val-boroPro)	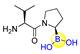	DPP4, FAP, DPP8, DPP9	IC_50_ of DPP4 < 4 nM; IC_50_ of FAP = 560 nM; IC_50_ of DPP8 = 4 nM; IC_50_ of DPP9 = 11 nM [36,37]; Phase II studies for prostate cancer and solid tumors; Phase I/II study for pancreatic cancer (https://adisinsight.springer.com/drugs/800016264, accessed on 5 November 2021).
**9**(ARI-3099)	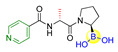	FAP	IC_50_ of FAP = 36 nM; inhibition of cellular prolyl endopeptidase activity but no cytotoxicity in murine FAP transfected HEK293 cells were observed [41,42].
**10**(ARI-3531)	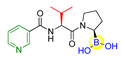	PREP	IC_50_ of PREP = 1.3 nM; complete inhibition of cellular prolyl endopeptidase activity in the mock-transfected cells but only approximately 60% inhibition in the mFAP transfected cells were observed [41,42].
**11**	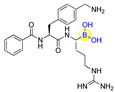	virus protease	IC_50_ of Dengue virus protease = 66 nM; IC_50_ of West Nile virus protease = 110 nM; IC_50_ of Zika virus protease = 250 nM; Antiviral effect: EC_50_ of Dengue virus = 30 μM; EC_50_ of West Nile virus = 38 μM [43].

SUB1 is a subtilisin-like serine protease that is critical during malaria parasite egress. To counteract the increasing resistance to available anti-malarial drugs, Lidumniece et al. identified a series of peptide boronic acids that could prevent malaria parasite replication and disease progression with high potency by targeting SUB1 [44]. In the study, a pentapeptidic α-ketoamide inhibitor was previously discovered with high potency, but unfortunately it showed no anti-parasite activity in vitro, probably due to poor cell permeability [45]. To address this poor cell permeability, the α-ketoamide functionality was replaced with a boronic acid warhead, and the prime side carboxylate was removed, giving the target compound **12**, which dramatically improved potency by about 7-fold over the best α-ketoamide (Figure 4) [44]. Further replacing P4 Ile residue to cyclopentane led to the low nanomolar IC_50_ compound **13** (Figure 4). Furthermore, the capacities of the compounds to interfere with parasite replication were assessed, with EC_50_ values of around 2 μM for **12** and **13**. The poor correlation between growth inhibition and the PfSUB1 enzyme-inhibitory potency might arise from the poor membrane penetration. Therefore, the P3 threonine residue of **13** was converted to valine, resulting in compound **14**, which showed gratifyingly enhanced growth inhibitory potency by ~10-fold (Figure 4). This was attributed to increased lipophilicity, which conferred better cellular permeability. Importantly, compound **14** was confirmed as a slowly reversible inhibitor against PfSUB1, showing a slow off rate due to the interaction between boronic acid and serine residues. This rendered compound **14** with an estimated bound half-life of ∼30 min, and thereby an appreciable pharmacodynamic property [44].

### 2.2. Threonine Protease Inhibitors

The proteasome represents a vital target for treating a series of diseases, including cancer, inflammation, and immune diseases. Since boronic acids are known to be more prone to covalently binding to threonine residues than to cysteine residues in the proteasome, peptide boronic acids have been identified as one significant class of covalent reversible proteasome inhibitors that display high selectivity.

Li’s group designed and synthesized six dipeptide proline-boronic acids and fourteen tripeptide proline-boronic acids as proteasome inhibitors, with the aim of overcoming the problems of metabolic instability and dose-limiting toxicities that exist in bortezomib and its analogues with a peptide backbone [46,47]. As a result, compound II-7 (**15**), which bears an N-terminal Cbz group, was demonstrated to be the strongest inhibitor with an IC_50_ value of 9.8 nM against human gastric carcinoma MGC803 cells (Figure 5). Further examination of the anti-tumor spectrum of II-7 showed comparable inhibitory activity against several different tumor cells lines with bortezomib (Figure 1 and Figure 5). II-7 was also confirmed to display high subunit selectivity towards the chymotrypsin-like (ChT-L) activity of the proteasome (IC_50_ = 18 nM), implying that it may have low off-target effects [47]. Furthermore, a novel class of urea-bearing peptide boronic acids were also designed and synthesized as potent proteasome inhibitors [48]. Further optimizations resulted in the cyclized 1,2,3,4-tetrahydroisoquinoline derivative I-14 (**16**) as the most potent compound (Figure 5). I-14 showed very strong inhibitory activity against proteasome ChT-L activity with an IC_50_ value below 1 pM and comparable potency against several different tumor cell lines as well as an improved pharmacokinetic profile when compared to bortezomib. Although I-14 exhibited slightly lower potency in inhibiting tumor growth in a Bel7404 mouse xenograft model than bortezomib, it demonstrated relatively low toxicity, suggesting I-14 as a promising drug candidate [48].

Lei et al. have developed a range of novel dipeptidyl boronic acid compounds, in which several compounds were demonstrated to strongly inhibit both β2 and β5 subunits of 20S proteasome and thus potently inhibit growth of both the multiple myeloma (MM) cancer cell lines and the solid tumor triple-negative breast cancer cell line MDA-MB-231 [49]. Additional in vivo pharmacokinetic results indicate that the prodrug 8t (**17**) exerts acceptable biological parameters with half-lives of 2.41 h for i.g. administration and 2.08 h for i.v. administration, as well as greater inhibition against the ChT-L activity of β5 subunit than the marketed MLN9708 in whole blood with the same dose at different time periods (Figure 5) [49]. It was concluded that 8t might be a promising candidate for the treatment of both MM and triple-negative breast cancer.

In addition to the specific inhibitors against proteasome as described above, Fang’s group recently developed a novel series of dual inhibitors against both the proteasome and HDAC by integrating the pharmacophores of HDAC inhibitors with bortezomib [50]. By introducing different zinc-binding groups to peptide boronates, hydroxamic acid derivatives and o-amino-benzamide derivatives with tuned amino acid residues were designed, synthesized, and subsequently assessed. Compound ZY-2 (**18**) was finally tested as the most potent dual inhibitor with excellent inhibition of the proteasome and an acceptable selectivity profile against HDAC1 (IC_50_ values of 1.1 and 255 nM, respectively) (Figure 5). Subsequently, antiproliferative activity of ZY-2 against the MM cell lines RPMI-8226, U266 and KM3, was confirmed with the IC_50_ values of 6.66, 4.31, and 10.1 nM, respectively. Interestingly, ZY-2 exhibited higher antiproliferative activity against the bortezomib-resistant MM cell line KM3/BTZ when compared to bortezomib (IC_50_ values of 8.98 vs. 226 nM) and was particularly more potent than the combination of MS-275 and bortezomib (1:1) (IC_50_ values of 8.98 vs. 98.0 nM) [50]. This study represents the first report to develop covalent reversible proteasome inhibitors with HDAC inhibitory activities for the potential treatment of bortezomib-resistant MM.

### 2.3. Aspartyl Protease Inhibitors

HIV-1 protease is an aspartic acid protease which is identified to be crucial for viral replication. As a competitive inhibitor of HIV-1 protease, darunavir has been approved by the Food and Drug Administration (FDA) and demonstrated to be highly efficacious in suppressing HIV-1 replication, dramatically contributing to the lifetime of HIV/AIDS patients [51]. Raines’s group recently replaced an aniline moiety in darunavir with a phenylboronic acid, resulting in a new derivative, **19**, with 20-fold greater affinity for the protease (Figure 6) [52]. X-ray crystallography confirmed that this boronic acid moiety makes three hydrogen bonds with HIV-1 protease, and even maintains its hydrogen bonds and its affinity toward the D30N variant of HIV-1 protease, which normally endows drug resistance. Furthermore, density functional theory analysis suggested that the short hydrogen bonds between the hydroxyl groups of the boronic acid and Asp30 (or Asn30) are covalent reversible bonds [52]. Although the antiviral activity in cell-based assays was assessed to be less potent than darunavir, possibly due to the low cell permeability, the boronic acid group might still be regarded as a versatile functional group in the design of small-molecule ligands [52,53].

### 2.4. Arginase Inhibitors

Arginases, a class of metalloproteases comprising two isoforms, ARG-1 and ARG-2, are responsible for converting L-arginine into L-ornithine and urea in the urea cycle (Figure 7A). The overexpression of arginase tends to lead to the depletion of arginine followed by suppression of T and NK cell growth, resulting in the tumor’s escape of the immune response [54]. Thus, it is expected that anti-tumor immune responses might be enhanced through the inhibition of arginase. A series of dipeptide boronic acids as arginase inhibitors were identified with the aim of improving the pharmacokinetic profile. Numidargistat (**20**) is a highly potent arginase inhibitor currently tested in phase 1/2 clinical trials for the treatment of patients with solid tumors malignancies (Figure 7B) [54]. Significant inhibition of human ARG-1 was recently observed for compounds **21**, **22**, and **23**, which exerted high potency with IC_50_ values of 0.8 nM, 1.6 nM and 0.8 nM, respectively (Figure 7B) [54].

## 3. Peptide Boronic Acids as Chemical Biology Tools

In addition to the formation of reversible covalent bonds to amino acids residing in proteases as described above, peptide boronic acids are also able to bind with diol units [5]. This diol recognition has made strategies for the recognition of glycans on proteins or cancer cell surface, delivery of RNA molecules, recognition of residues such as lysine-modified phosphatidylglycerol (Lys-PG) on bacterial cell surface, and even the development of pH devices feasible.

### 3.1. Recognition of Glycans on Proteins or Cancer Cell Surface

Glycans of membrane-bound and secreted proteins in healthy cells play important roles in cell signaling and mediating cell–cell interactions, as well as protecting proteins from degradation. However, specific glycans, such as sialyl Lewis X (sLex), have been found to be overexpressed on a variety of cancerous cells, and correlated with the cancer differentiation and metastasis [55]. Both antibodies and natural lectins have been proven valuable for sensing the glycans, but they also characterized by instability, high costs, and difficult production [56]. Peptide boronic acids represent the alternative molecules capable of selectively recognizing glycans on proteins and cell surface.

Thompson’s group described the design and synthesis of a combinatorial library of synthetic lectins (SLs) that incorporate boronic acids with the general sequence Ac-RXD*XXXD*XBBRM-resin, termed Fixed Position Library (FPL) (Figure 8A) [56], where D* denotes phenyl boronic acid (PBA)-functionalized diaminobutanoic acid (Dab), B indicates beta-alanine, and X is a randomized amino acid. In the library, arginine was introduced at the N-terminus for the favorable interactions with the negative charge present on sialylated glycoproteins. It was expected that this library covered a diversity of ∼1.6 × 10^5^ unique molecules. This library was then screened against several FITC-labeled glycoproteins bearing unique glycans, including ovalbumin (OVA), bovine submaxillary mucin (BSM), and porcine stomach mucin (PSM). Finally, the synthetic lectin SL2 (Figure 8A) was demonstrated to bind OVA with ∼3-fold selectivity over BSM and ∼5-fold selectivity over PSM. A derivative of SL2, termed SL2-Ala, which has alanine in place of the boronic acid-functionalized Dab moieties, was found not to bind with OVA, suggesting the critical role of boronic acid. This study has provided an approach for identifying novel SLs that are capable of selectively recognizing glycoproteins and potentially applicable for cancer diagnostics [56].

Zhang’s group prepared a borono-peptide (B(OH)_2_AEAEAELRARARL-OH) bearing the phenylboronic acid moiety followed by fabrication and self-assembly by mixing the borono-peptide with alizarin red S (ARS) to produce a unique peptide nanofibrous indicator (NFI) (Figure 8B) [57]. When encountering saccharides, the eye-detectable fluorescence change could be induced from its coordination state between ARS and the boronic acid group to its free state. Importantly, the NFI tends to avoid the cellular uptake, owing to the micrometer lengths and nanometer diameters. This has made it feasible to specifically recognize and discriminate between human hepatocellular liver carcinoma (HepG2) cells through direct naked-eye judgment [57]. In addition, they also reported boronic acid-functionalized peptide-based fluorescent sensors (BPFSs) that are able to recognize and discriminate cancer-associated glycans as well as cancer cell lines (Figure 8C) [58]. When the boronic acid of BPFSs binds to saccharides, the simultaneous formation of the B-N bond tends to confine the nitrogen lone-pair electrons on the amino group, resulting in the increase in anthracene fluorescence. By screening BPFSs with the altered peptide sequence and length, a fluorescent sensor BPFS1 bearing an FRGDF peptide was identified to specifically recognize cell-surface sLex and fluorescently label and discriminate between cancer cells with the aid of the interaction between the RGD sequence and integrins (Figure 8C) [58]. These BPFSs have great potential as fluorescent probes for clinical cancer diagnosis or therapy.

### 3.2. RNA Carriers or Binders

To deliver small interfering RNA (siRNA), Kataoka’s lab first prepared PEG-*b*-PLys, a platform cationic polymer consisting of poly(ethylene glycol)-block-poly(L-lysine) where 3-fluoro-4-carboxyphenylboronic acid (F-cPBA) was conjugated to the side chain of the L-lysines (Figure 9A) [59]. The resultant PEG-*b*-P(Lys/F-cPBA_23_)_42_, with 23 F-cPBA moieties, exhibited acceptable solubility and a stable complex with siRNAs under a physiological salt condition regardless of the presence of fetal bovine serum (FBS). Interestingly, the complex could be markedly destabilized upon addition of adenosine triphosphate (ATP) or uridine monophosphate (UMP) with ribose moieties at their in vivo concentrations, making it feasible to programmatically release siRNAs upon intracellular arrival (Figure 9A) [59]. It should be noted that the number of F-cPBA was also demonstrated to be critical for the solubility and stability of the complex.

Santos’s group generated a branched peptide (BP) library bearing boronic acids (46,656 compounds) by split and pool synthesis and carried out on-bead high throughput screening against HIV-1 Rev response element IIB (RRE IIB) (Figure 9B) [60]. Three peptides (BPBA1, BPBA2, and BPBA3) showed low micromolar binding affinities with K_D_ values of 1.4, 3.3, and 8.7 μM, respectively. Removing the boronic acid moieties in the para position of BPBA1 reduced its binding against RRE IIB by 6-fold, which suggests the important role of the boronic acids. Further introduction of fluorine in the ortho position of BPBA1 contributes to a stronger interaction with RRE. This supports the idea that manipulating the Lewis acidity of the boronic acid could help tune the binding affinity of BPBAs [60]. In the following studies, the branching of BPBAs was confirmed to notably contribute to the binding affinity against RRE IIB. Acceptable selectivity of BPBAs towards RRE IIB wild-type structure was also observed, regardless of the presence of RRE IIB-related structural variant RNAs, competitor tRNAs, or an RRE IIB DNA analogue. Although BPBA1 exerted the strongest binding affinity, acceptable cell permeability, and minimal cytotoxicity in two eukaryotic cell lines, it failed to inhibit HIV-1 p24 capsid production [61]. Soon after, BPBA3 was finally demonstrated to significantly inhibit HIV-1 p24 capsid production in a dose-dependent manner with an IC_50_ value around 5 μM. It was suggested that BPBA3 could induce a conformational change of the tertiary structure in the internal loop to expose the RNA to RNase cleavage, thereby resulting in the inhibition of HIV-1 replication [62].

### 3.3. Recognition of Bacterial Cell Surfaces

Owing to the growing resistance threat of bacterial pathogens, there is a high demand for the development of novel strategies for infection diagnosis and antibiotic discovery. Bacterial cell surfaces have been found to display highly enriched phosphatidylethanolamine (PE) and lysine-modified phosphatidylglycerol (Lys-PG) for gaining resistance against cationic antibiotics [63]. Thus, the specific recognition of these bacterial lipids may give rise to novel diagnostic methods for bacterial infection, as well as solutions to the antibiotic-resistance problem. Against this backdrop, Gao’s group described an unnatural amino acid (AB1) bearing boronic acid that could selectively modify the lipids of interest, PE and Lys-PG, via reversible covalent bond formation mediated by iminoboronate chemistry under physiological conditions [63]. The control molecule without boronic acid was not able to label PE or Lys-PG, indicating the critical importance of the boronic acid moiety. Furthermore, AB1 was investigated to selectively stain Gram-positive bacteria bearing PE and/or Lys-PG as the major lipids on the cell membranes, in contrast with the inability to label Gram-negative *E. coli* and mammalian cells due to the lack of PE or Lys-PG on their membranes. To overcome the competitive reactions from lysine and lysyl residues of various proteins, AB1 was conjugated with a polycationic peptide Hlys with the sequence of RYWVAWRNR to give Hlys-AB1 (Figure 10A). It was found that Hlys-AB1 exhibited synergy between AB1 and Hlys and selectively labelled *S. aureus* cells at nanomolar concentrations even in the presence of 10% FBS, with little or no staining of mammalian cells and Gram-negative bacteria [63]. Based on its high bacterial selectivity and potency, Hlys-AB1 may represent a potentially useful tool for targeting the bacteria in blood serum or living organisms.

Furthermore, Gao and coworkers constructed a phage display library of peptides bearing two 2-acetylphenyl-boronic acid (APBA) moieties capable of reversible covalent binding to the biological amines on the bacterial cell surface via the dynamic formation of iminoboronates (Figure 10B) [64]. Panning the library directly against the live bacterial cells, including *Staphylococcus aureus* and *Acinetobacter baumannii*, provided submicromolar and highly selective peptide binders. Intriguingly, parallel screening by a control library that was unmodified by APBA failed to give potent peptide binders, demonstrating the integral nature of the APBA modifications. Furthermore, the identified peptides were found to be a promising delivery vehicle for eosin, a phototoxin that can trigger the production of reactive oxygen species (ROS) and subsequently kills cells in close proximity upon photoirradiation [64]. Importantly, little or no mammalian cell toxicity was observed with or without photoirradiation. It should be noted that the phage displayed peptides bearing APBA groups could reach up to 10^9^ unique peptide sequences, offering huge diversity and the capacity to screen peptide boronic acids. These results clearly demonstrate the feasibility of discovering species-specific antibiotics using an APBA-modified peptide library and paved a facile strategy for the discovery of novel antibiotics in future.

### 3.4. pH Responsive Devices

Peptide boronic acids can also be used to develop pH-responsive devices. Koksch’s group devised and characterized a novel polypeptide incorporating a phenylboronic acid derivative, BA-H5-St, enabling formation of a stable coiled coil structure at endosomal pH values (pH 5.0 to 5.5) [65]. The vesicle anchoring BA-H5-St could form an intervesicular complex with a sugar-like compound (phosphatidylinositol: PI)-containing liposome but would fail to achieve membrane fusion at physiological pH (pH 7.4). When reducing the pH from 7.4 to 5.0, BA-H5-St tended to form a coiled structure (endosomal condition). This shortened the distance between the pilot and target vesicles and thus promoted membrane fusion. Importantly, the boronic acid moiety was confirmed to be crucial for the pH-responsive membrane fusion behavior by recognizing the cis-diol group on the target liposome (Figure 11A) [65].

Gao’s group developed a novel strategy for peptide cyclization in which intramolecular iminoboronate formation allowed spontaneous cyclization under physiologic conditions to yield monocyclic or bicyclic peptides (Figure 11B) [66]. It was important that the iminoboronate-based cyclization could be rapidly reversed in response to multiple stimuli, such as pH values [66]. This highly versatile strategy for peptide cyclization has promising applications in many areas of chemical biology.

## 4. Boronic Acid-Mediated Peptide Synthesis and Modifications

While boronic acids can be incorporated into peptides for the development of a series of covalent enzyme inhibitors and chemical biology tools, boronic acids have also been demonstrated to be promising reagents for peptide synthesis and modifications. Not only the peptide cyclization and dipeptide synthesis, boronic acids can also mediate the peptide modifications at N-terminal, side chain, and backbone positions.

### 4.1. Boronic Acid-Mediated Peptide Cyclization and Dipeptide Synthesis

Natural products such as biphenomycins and arylomycins have been found to contain biaryl peptide motifs [67]. Afonso et al. attempted to apply the Miyaura borylation on resin for the solid-phase synthesis of biaryl-bridged macrocyclic peptides via an intramolecular Suzuki–Miyaura cross-coupling reaction [68]. During the synthesis, the boronated peptides were first synthesized using Miyaura borylation followed by attaching a halogenated derivative of phenylalanine. The resultant linear peptidyl resin precursor, containing both the boronate and halogenated aromatic amino acid, was subjected to Suzuki–Miyaura macrocyclization under microwave irradiation, leading to biaryl cyclic peptides with different ring sizes (Figure 12A) [68]. James’s group also conducted solid-phase Suzuki–Miyaura macrocyclizations and achieved the synthesis of a range of biaryl-bridged macrocyclic peptides, which possess not only meta-meta, but also meta-ortho and ortho-meta configurations by tuning the regiochemical variation of the aryl boronate and aryl halide coupling partners (Figure 12A) [69]. This study was complementary to the para-para and meta-para systems realized by Afonso et al. and might allow for the fine-tuning of macrocyclic peptide conformation. In the same manner, Kemker et al. recently established a robust method for side chain-to-tail cyclization on resin between the indole ring of bromotryptophan and an N-terminal boronic acid through Suzuki−Miyaura cross-coupling. This led to the macrocyclized peptides with good selectivity and high affinities towards αVβ3, as well as high plasma stability (Figure 12B) [70]. Moreover, Chan–Evans–Lam coupling, mediated by borylated amino acids, was also utilized as a key step for the effective assembly of the isodityrosine subunit during the total synthesis of antiallergic depsipeptide seongsanamide A [71]. All of the aforementioned studies have provided straightforward approaches for efficient peptide cyclization, which may be potentially applicable for natural products’ synthesis and the construction of macrocyclic peptide libraries.

Apart from the peptide macrocyclization, boronic acid has also been utilized for catalyzing dipeptide synthesis. Blanchet’s group described a catalytic dipeptide synthesis using (2-(thiophen-2-ylmethyl)phenyl)boronic acid **24** as a catalyst (Figure 12C) [72]. Taking a step forward, a 2-chloro-substituted biaryl boronic acid **25** was observed efficiently catalyzing the peptide bond condensations, giving fourteen desired dipeptides in good yields with little to no racemization (Figure 12C) [73]. Takemoto’s group devised an ortho-hydroxyphenyl-substituted gem-diboronic acid **26** capable of catalyzing peptide condensations (Figure 12C) [74]. It was found that the gem-diboronic acid catalyst could efficiently activate the carboxy group of α-amino acids by forming bidentate intermediates, which then led to amidation by nucleophilic amino acids to afford the corresponding dipeptides. Interestingly, the catalyst was compatible with α-amino acids bearing several different N-protecting groups, such as Boc, Cbz, Fmoc, and N-trifluoroacetyl, and activated their carboxy groups to furnish the desirable dipeptides in good yields with low epimerization [74]. Moreover, nucleophilic amino acids endowed with a variety of functionalized side chains including phenyl, alkyne, sulfide, ester, amide, and imidazole were also tolerated, which implies the versatility of this strategy. This would open new avenues for catalytic peptide condensation under mild conditions.

### 4.2. Boronic Acid-Mediated Peptide Modifications

Peptide modifications represent an alternative way to modulate the interaction with the target, improve stability and solubility, anchoring useful reactive handles for conjugation, imaging, or stapling, etc. Based on its acceptable aqueous stability and solubility, boronic acid has shown promise in this field specifically. Mediated by a variety of boronic acids, diverse modifications at different positions, including N-terminus, side chain, and backbone, have been realized.

#### 4.2.1. N-Terminal Modifications

For the peptide modifications at N-terminus, 2-formylphenylboronic acid (2-FPBA) was surveyed to be versatile, leading to many applications. Gao’s group reported that 2-FPBA and L-2,3-diaminopropionic acid (L-Dap) tend to form a rapidly reversible and chemoselective imidazolidino boronate (IzB) complex (Figure 13A) [75]. The IzB complex formation readily proceeds in biological milieu and specifically responds to cysteine even in the presence of serine, glucose, lysine, or glutathione. This enables the generation of IzB-cyclized peptides to detect free cysteine and, on the basis of that, they reported the dynamic change of cysteine in complex biological media (Figure 13A) [75]. A 2-FPBA derivative was also developed and applied for intramolecular acyl transfer upon the formation of a TzB intermediate, producing N-acylthiazolidines with robust stability under physiological conditions (Figure 13B) [76]. Gois’s group described a fast and reversible conjugation of 2-FPBA with N-terminal cysteines (NCys) to afford a thiazolidine boronate (TzB) complex, enabling a facile modification of a NCys-bearing peptide or proteins in complex biological milieu (Figure 13C) [77]. Based on the above study, Luk’s group coupled 2-FPBA with asparaginyl endopeptidases (AEPs) ligation for the efficient N- and C-terminal ligation of polypeptides or proteins (Figure 13D) [78]. When AEP can efficiently hydrolyze the C-terminal amide bond of a Asn–Cys–Leu sequence and subsequently mediate ligation to the N-terminus of an incoming nucleophile peptide, the byproduct of the enzymatic reaction Cys–Leu could be selectively trapped by adding 2-FPBA (Figure 13D) [78]. Interestingly, 2-FPBA, as a scavenger, could significantly improve the ligation yield by over 40%. In a different way, the N-terminal cysteine can be firstly conjugated with a boron hot-spot (BHS) based on the 3-hydroxyquinolin-2(1H)-one scaffold to enable direction of 2-FPBA carbonyl function towards the N-terminal amino group (Figure 13E) [79]. Importantly, this BHS was confirmed to be selective in the presence of competing lysine amino groups in peptides and prone to forming the N-terminal iminoboronate instead of side chain iminoboronates. Additionally, the resultant iminoboronates exhibited acceptable stability in buffer solutions and were cleavable in the presence of glutathione. Furthermore, the approach was applied for the preparation of cleavable fluorescent conjugates with a laminin fragment, allowing it to target the 67LR receptor and delivering cargo to cancer HT29 cells [79].

Unlike the N-terminal cysteine modification using 2-FPBA, several other approaches mediated by boronic acids reagents for N-terminal modifications were also developed. Ball’s group also described an ascorbic acid-mediated N-terminal functionalization of peptides in air under mild conditions (room temperature, pH 7) with the vinylboronate reagents (Figure 13F) [80]. Using angiotensin I (H-DRVYIHPFHL-OH) as model peptide, they observed that a variety of aryl- and alkyl-substituted vinylboronic acids could be utilized to selectively modify the N-terminus in appreciable yields (Figure 13F). Furthermore, the reaction was compatible with a wide range of amino acids at the N-terminal position (e.g., Val, Met, Trp, Cys, His, Ser, Tyr, and Arg) [80]. In addition, the ortho-sulfonamide arylboronic acids could be utilized for site-selective arylation at the N-terminus. This reaction proceeds under mild conditions in water and is compatible with a wide range of N-terminal residues (Figure 13G) [81]. These studies have provided convenient, mild, and selective approaches for single modification at the N-terminus of peptides.

#### 4.2.2. Side-Chain Modifications

With respect to side-chain modification, Ball’s group developed a nickel (II) salt-promoted cysteine arylation on peptides using 2-nitro-substituted arylboronic acids reagents (Figure 14A) [82]. The reaction is rapid and selective towards the side chain of cysteine residues under physiological conditions in purely aqueous media. It was further confirmed by investigating a series of cysteine-containing peptides, resulting in the desired arylated-cysteine-containing peptides in appreciable yields even in the presence of competitive nucleophilic residues [82].

Anslyn’s group demonstrated that 2-FPBA was able to afford an irreversible three-component complex in aqueous media with catechol and N-hydroxylamines [83]. This complex, once formed, was confirmed to be stable under neutral aqueous conditions for more than 72 h. Intriguingly, the complex exhibited excellent orthogonality for common biological functional groups and other bioorthogonal coupling reactions including copper(I)-catalyzed alkyne-azide cycloaddition and aminoether/carbonyl condensations. This enables the dual-labelling of a peptide containing L-dihydroxyphenylalanine (L-DOPA) in one pot (Figure 14B) [83].

Ricardo et al. described the peptide side-chain derivatizations on-resin at Lysine and *N^ε^*-Me-Lysine residues by means of the Petasis (Borono¬–Mannich) reaction (Figure 14C) [84]. By varying the carbonyl and boronic acid components, it allowed the simultaneous peptide side-chain modifications with diverse aryl moieties and sugars. In addition, solution-phase peptide macrocyclization could also be realized with high efficiency using the versatile Petasis reaction, which is capable of installing different aryl tethers and additional derivatizations from the carbonyl component [84].

Kondo et al. reported an efficient methodology for the direct radiolabeling of a peptide side chain via copper-mediated radioiodination by using an aryl boronic acid precursor through a Chan–Evans–Lam cross-coupling reaction (Figure 14D) [85]. This approach allows for the avoidance of heavy metals, heating, oxidizing reagents, an inert condition, as well as the preparation of complex starting materials. In addition, it can be carried out at room temperature, and is easily separated due to the significantly different polarity between the boronic acid precursor and the iodine compound [85].

#### 4.2.3. Backbone Modifications

With respect to the backbone modifications, Ball’s group provided a convenient approach for accessing the N-alkenyl or N-aryl derivatives of complex peptide by histidine-directed alkenylation/arylation of backbone N-H bonds mediated by copper (II) salt in the presence of boronate reagents (Figure 15A) [86]. It was found that the backbone N−H bond preceding the histidine of thyrotropin-releasing hormone (TRH) could be specifically arylated with arylboronic acid at ambient temperature in the presence of copper (II) salts. Copper (II) and histidine were demonstrated to be crucial, and it was speculated that copper could coordinate to the imidazole side chain of histidine and activate N−H bonds preceding histidine. Leuprolide, a histidine-containing peptide also bearing N-terminal pyroglutamate (Glp) residue, could also be effectively modified with a broad range of arylboronates and alkenylboronates. Moreover, angiotensin I was tested and determined to easily react with alkenylboronates, but it showed limited arylboronate reactivity. This is possibly due to the steric hindrance of the aryl reagents relative to alkenyl reagent [86].

Further, Ball’s group applied the backbone N−H alkenylation for a photocleavable backbone N−H modification, which allowed the efficient photocaging of folding and function of peptides or proteins (Figure 15B) [87]. A variety of peptides, including angiotensin I, a chymotrypsin substrate peptide (CSP), leuprolide, and a collagen mimetic peptide (CMP), were chosen and evaluated for the photocaging process. Clearly, the peptides could be converted to backbone-modified products using boronic acid reagents in the presence of a copper (II) salt and regenerated to the uncaged peptides upon irradiation with 365 nm light (Figure 15B) [87]. Very recently, they also developed a molecule with two boronic acid functional groups, which could be subjected to nickel (II) salt-catalyzed cysteine arylation followed by histidine-directed alkenylation mediated by copper (II) salt (Figure 15C). This enables the preparation of stapled peptides, peptide-protein conjugation, etc. [88].

## 5. Chemical Preparations of Peptide Boronic Acids

Owing to the versatile and broad applications of peptide boronic acids, several simple and practical accesses were realized, making it possible to achieve a variety of peptide boronic acids that can be potentially applied in drug discovery, life sciences, and materials research.

By employing the strong complex between boronic acid and glycerol, Behnam et al. applied a commercially available 1-glycerol polystyrene resin for solid-phase peptide synthesis (SPPS) of peptide boronic acids in a high yield [89]. The approach was compatible with standard Fmoc chemistry and extendable to aryl and alkyl aminoboronic acids and even peptidosulfonamides. The resultant peptide boronic acids could be efficiently released from the resin, leading to straightforward access to peptide boronic acids (Figure 16A). Moreover, orthogonally protected intermediates with boronic acid at the C-terminus could be easily obtained [89].

Yudin’s lab showed the use of α-boryl aldehydes in the synthesis of β-aminoboronic acid derivatives [90]. They used a N-methyliminodiacetyl boronates group to explore polar reactivity in the vicinity of boron, and the N-methyliminodiacetyl group was removed in the presence of acid. Thus, the β-aminoboronic acids were synthesized successfully in this manner. This mild method allowed researchers to develop a new approach to β-aminoboronic acid derivatives. In addition to solution phase synthesis, Yudin’s lab applied a similar approach to transform the N-terminus of a linear peptide into a β-aminoboronic acid on solid support (Figure 16B) [91,92]. This novel method is completely compatible with traditional SPPS, as only the final compound needs to be purified, and it also allows chemists to generate peptide libraries with aminoboronic acids for biological purposes.

Alkyl carboxylic acids are present in a wide range of natural products and medicines. Baran’s group recently realized a nickel-catalyzed decarboxylative borylation by using B2pin2 (Bpin = pinacol boronate) as a boron source, directly converting carboxylic acids into boronate esters (Figure 16C) [93]. Importantly, this reaction was reported to be compatible with a variety of functional groups, such as alkyl/aryl halides, amides/carbamates, alcohols, ketones, and olefins. In this manner, not only a series of α-amino boronic acids but also the alkyl boronic acid drugs Velcade and Ninlaro, as well as a boronic acid analogue of vancomycin, were achieved in good yields [93]. By means of this approach, twenty Fmoc-protected natural amino acids with orthogonal side-chain protections were first activated through the formation of N-hydroxyphthalimide (NHPI) esters, which subsequently underwent Ni-catalyzed decarboxylative borylations to give N-Fmoc-α-pinacolyl boronates (Figure 16D) [94]. The free boronic acids could be achieved after the deprotection of pinacolyl building blocks through the monophasic transesterification method using volatile methylboronic acid [95]. Furthermore, the resultant Fmoc-α-aminoboronates could be immobilized onto commercially available 1-glycerol polystyrene resin followed by standard Fmoc SPPS to yield a diversity of peptide boronic acids with high purity [94].

As a last example in this section, Roelfes’s group reported a rapid and selective copper II-catalyzed β-borylation of dehydroalanine (Dha) residues in ribosomally synthesized and post-translationally modified peptides (RiPPs) under mild conditions (Figure 16E) [96]. Three RiPPs, including thiostrepton, nosiheptide, and lanthipeptide nisin Z, were efficiently borylated with high selectivity by using copper II-catalysis under mild conditions. It was observed that the borylation of thiostrepton provided significantly improved water solubility, with value that were 84 times greater than previously. The antimicrobial activities of thiostrepton and nosiheptide were preserved, as confirmed by minimum inhibitory concentration (MIC) assays. Furthermore, the introduction of boronic-acid functionalities was proven to be valuable, and versatile motifs for further chemical transformations, such as turn-on fluorescent and pH-controlled reversible labeling of RiPPs [96].

## 6. Conclusions

This review summarizes the latest advances on the broad applications of peptide boronic acids in the past ten years. The identification of covalent reversible enzyme inhibitors, the recognition and detection of glycans on proteins and cancer cell surfaces, delivery of siRNAs, developing pH-responsive devices, recognition of RNA, as well as bacterial surfaces, were included in this review. Particularly, the highly diverse peptide boronic acid library prepared by phage-display or split and pool synthesis could potentially be used for discovering novel peptide boronic acids in future. We discussed boronic acids as attractive reagents for realizing chemical peptide macrocyclization by Suzuki–Miyaura cross-coupling reaction, catalytic dipeptide synthesis, diverse modifications of peptides at the N-terminal, side chains and backbone. Additionally, the facile and convenient chemical preparations of peptide boronic acids could also pave the way for preparing a growing number of peptide boronic acids that could potentially be applied in medicinal chemistry, chemical biology, and material sciences. All in all, boronic acid has been demonstrated to be a versatile and robust functional group that is widely integrated with peptides, marrying the best of boronic acid and peptide. It is anticipated that wider research interests will be provoked in this field due to its enormous potential.

## Figures and Tables

**Figure 1 ijms-22-12958-f001:**
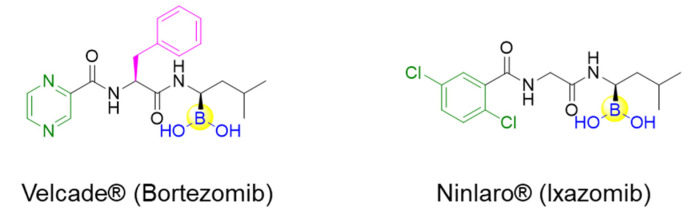
Structures of the approved peptide boronic acid medicines.

**Figure 2 ijms-22-12958-f002:**
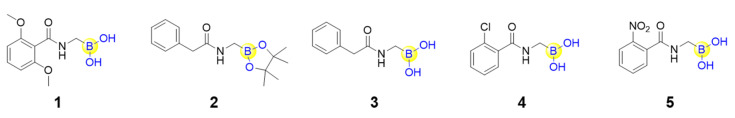
Structures of the five peptide boronic acids binding to a penicillin-binding protein.

**Figure 3 ijms-22-12958-f003:**
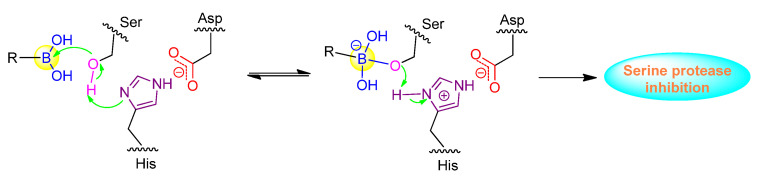
Inhibition mechanism of peptide boronic acids against serine protease.

**Figure 4 ijms-22-12958-f004:**
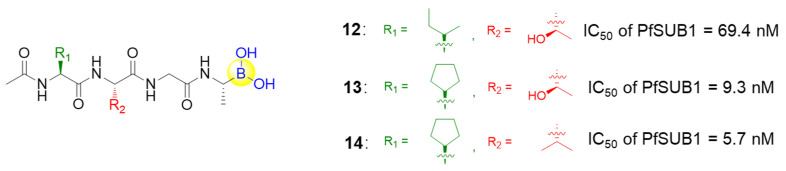
Representative peptide boronic acids as SUB1 inhibitors.

**Figure 5 ijms-22-12958-f005:**
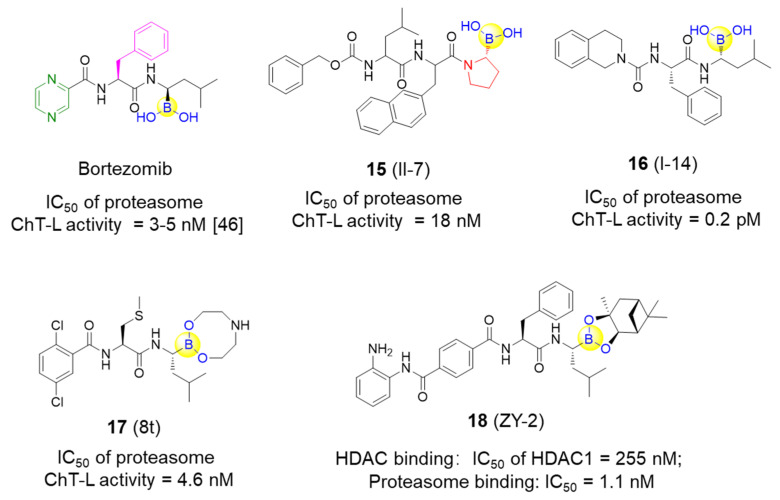
Representative peptide boronic acids have been used as threonine protease inhibitors since 2011.

**Figure 6 ijms-22-12958-f006:**
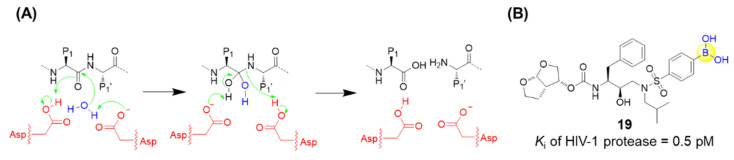
(**A**) Catalytic site of HIV-1 protease; (**B**) Structure of compound **19**.

**Figure 7 ijms-22-12958-f007:**
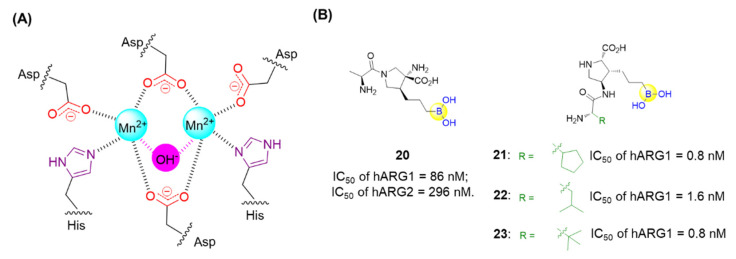
(**A**) Catalytic site of hARG-1; (**B**) Representative peptide boronic acids have been used as arginase inhibitors since 2011.

**Figure 8 ijms-22-12958-f008:**
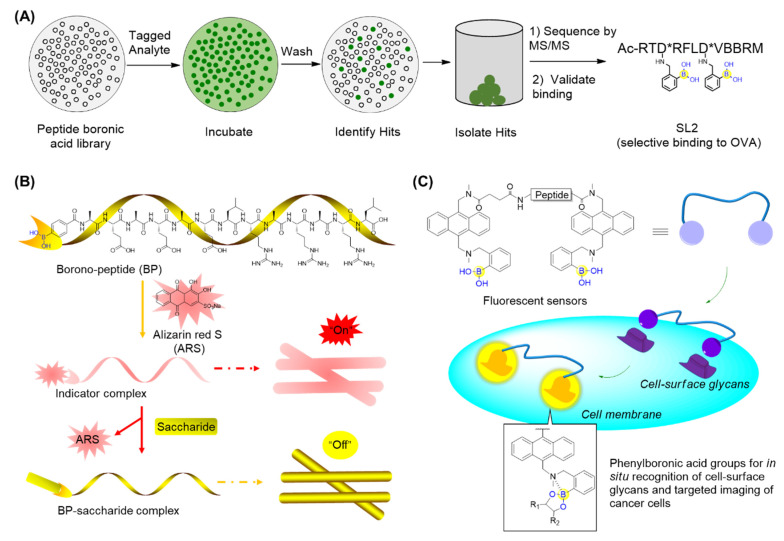
Recognition of glycans using peptide boronic acids: (**A**) Identification of boronic acid functionalized peptidyl synthetic lectins (SLs) using a combinatorial library; (**B**) The borono-peptide (BP) and BP-induced nanofibrous indicator (NFI) for cancer cell detection; when encountering saccharides on cancer cell surfaces, the eye-detectable fluorescence change could be induced from its coordination state between ARS and boronic acid group to its free state; (**C**) Boronic acid-functionalized peptide-based fluorescent sensors that are able to recognize and discriminate the cancer-associated glycans as well as cancer cell lines.

**Figure 9 ijms-22-12958-f009:**
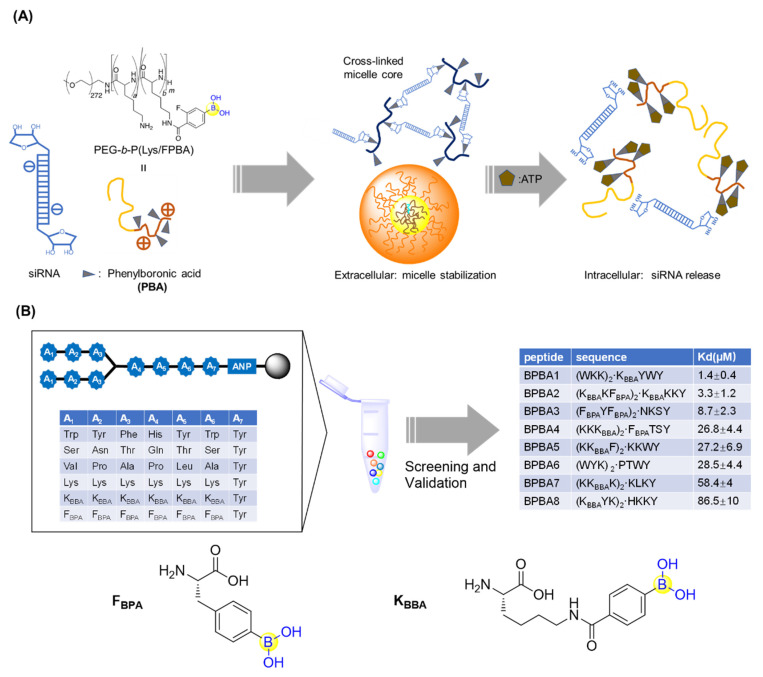
(**A**) Phenylboronic acid-functionalized peptide for siRNAs delivery; (**B**) Screening of branched peptide boric acids targeting RNA.

**Figure 10 ijms-22-12958-f010:**
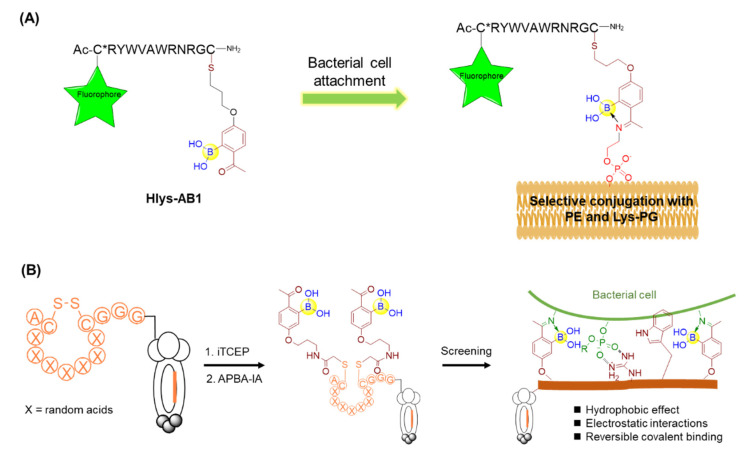
(**A**) Covalent recognition of membrane lipids using peptide boronic acids; (**B**) Phage display of iminoboronate-modified peptide library for screening against bacterial cell.

**Figure 11 ijms-22-12958-f011:**
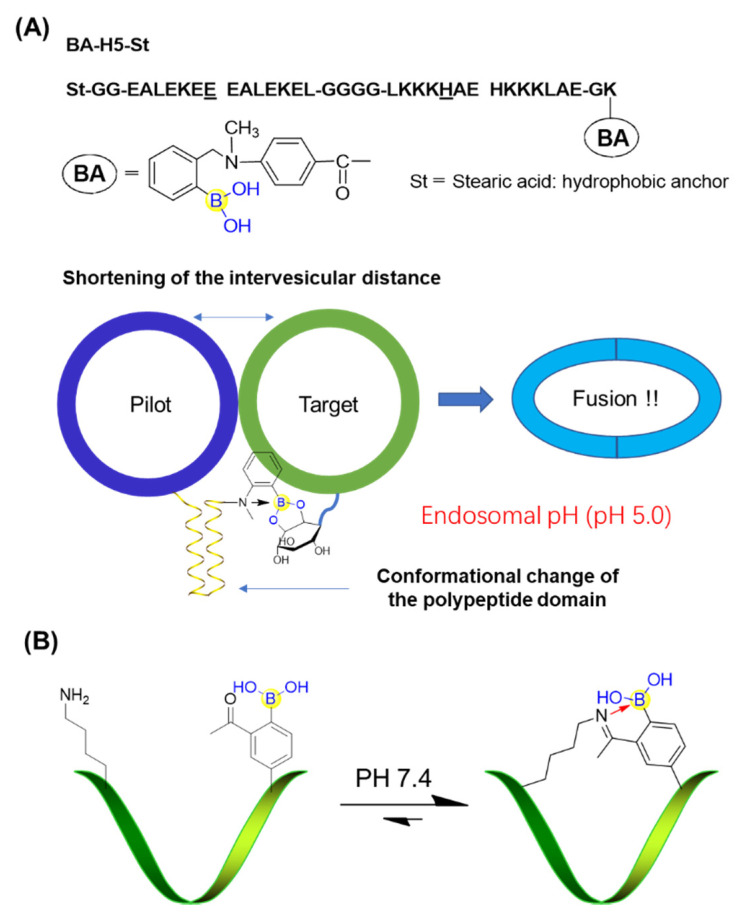
(**A**) pH-responsive membrane fusion system based on peptide boronic acid; (**B**) Iminoboronate-cyclized peptide responding to pH stimuli.

**Figure 12 ijms-22-12958-f012:**
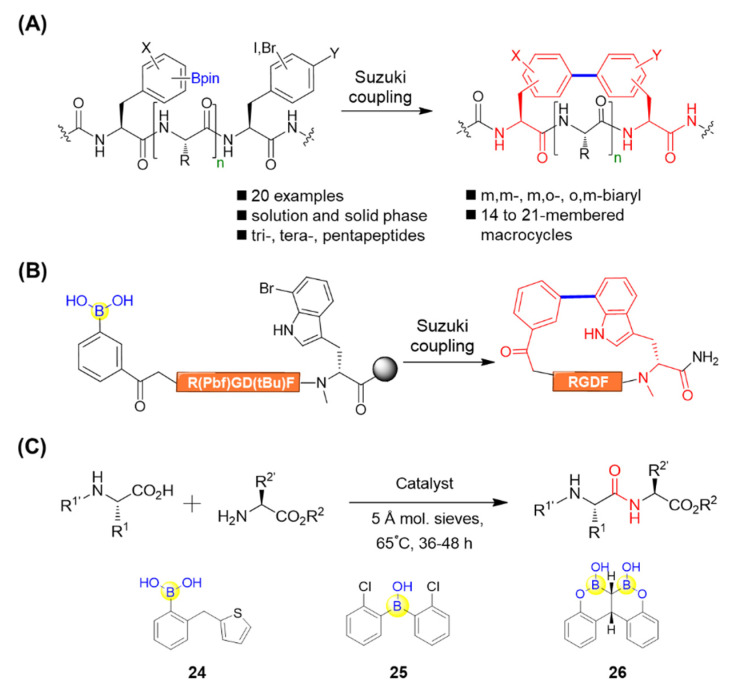
(**A**) Suzuki–Miyaura macrocyclization for biaryl cyclic peptide synthesis mediated by boronic acid; (**B**) Suzuki−Miyaura macrocyclization of RGD peptides mediated by boronic acid; (**C**) Boronic acid molecules as catalysts for dipeptide synthesis.

**Figure 13 ijms-22-12958-f013:**
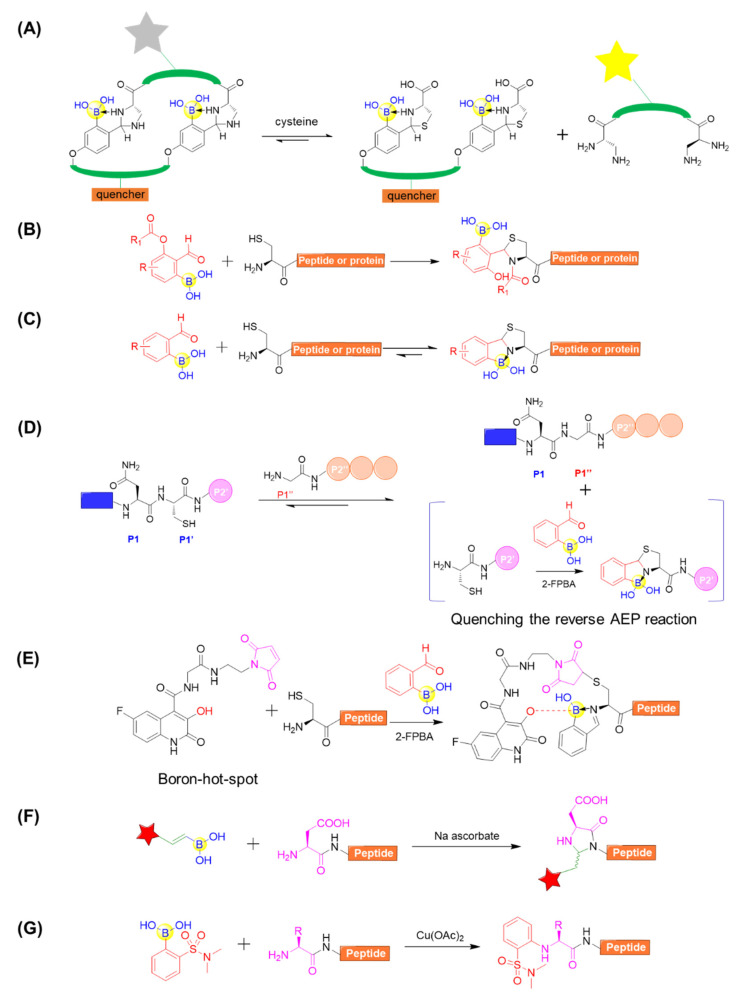
(**A**) Cysteine detection using chemoselective imidazolidino boronate (IzB)-cyclized peptide; (**B**) N-Terminal cysteine modification through TzB mediated acyl transfer reaction; (**C**) Conjugation of N-terminal cysteine (NCys) with 2-formylphenylboronic acid (2-FPBA) to afford a thiazolidine boronate (TzB) complex; (**D**) Coupling of AEP ligation with N-terminal cysteine trapping by 2-FPBA; (**E**) Site-selectively installing an iminoboronate using a boron hot-spot (BHS) strategy; (**F**) Modification of peptide using alkenyl boronic acid and sodium ascorbate; (**G**) Copper-mediated site-selective peptide arylation at the N-terminus.

**Figure 14 ijms-22-12958-f014:**
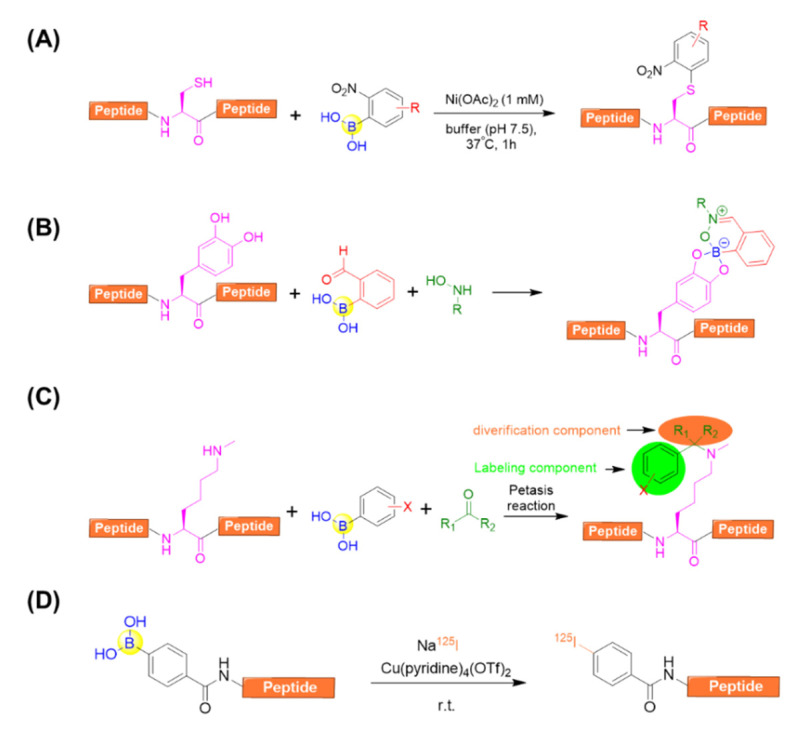
(**A**) A nickel (II) salt-promoted cysteine arylation on peptides using 2-nitro-substituted arylboronic acids reagents; (**B**) A three-component assembly for the purpose of labeling peptides containing L-dihydroxyphenylalanine (L-DOPA) in one pot; (**C**) Peptide side-chain derivatizations N^ε^-Me-Lysine residue by means of the Petasis (Borono–Mannich) reaction; (**D**) Direct radiolabeling of peptides via copper-mediated radioiodination using an aryl boronic acid precursor.

**Figure 15 ijms-22-12958-f015:**
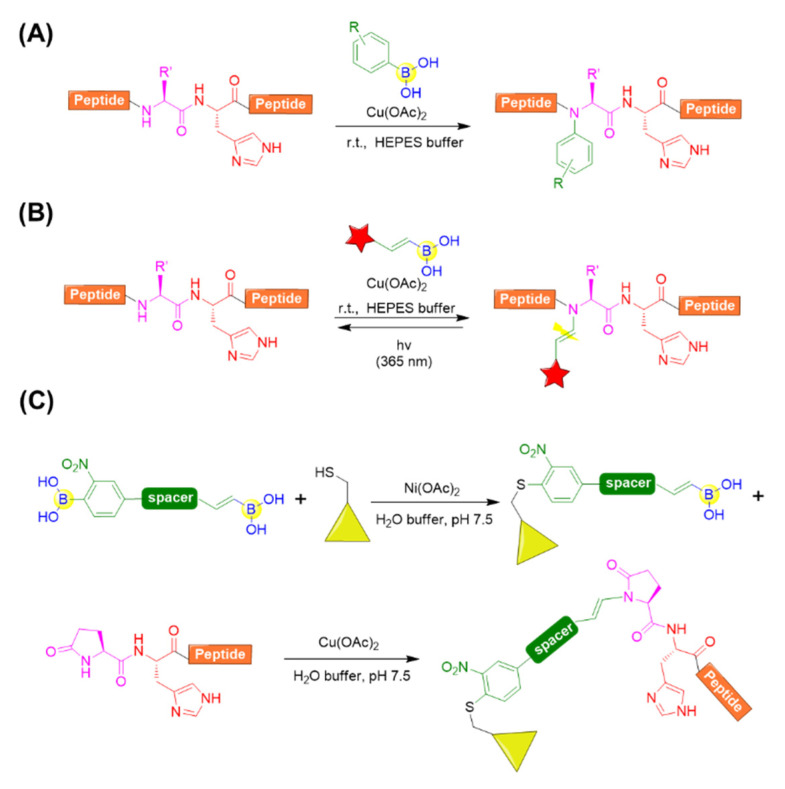
(**A**) Histidine-directed arylation of backbone N-H bonds mediated by copper (II) salt in the presence of boronate reagents; (**B**) Vinylogous photocleavable modification of backbone N−H bonds by histidine-directed alkenylation; (**C**) Molecule with two boronic acid functional groups for sequential bioconjugation.

**Figure 16 ijms-22-12958-f016:**
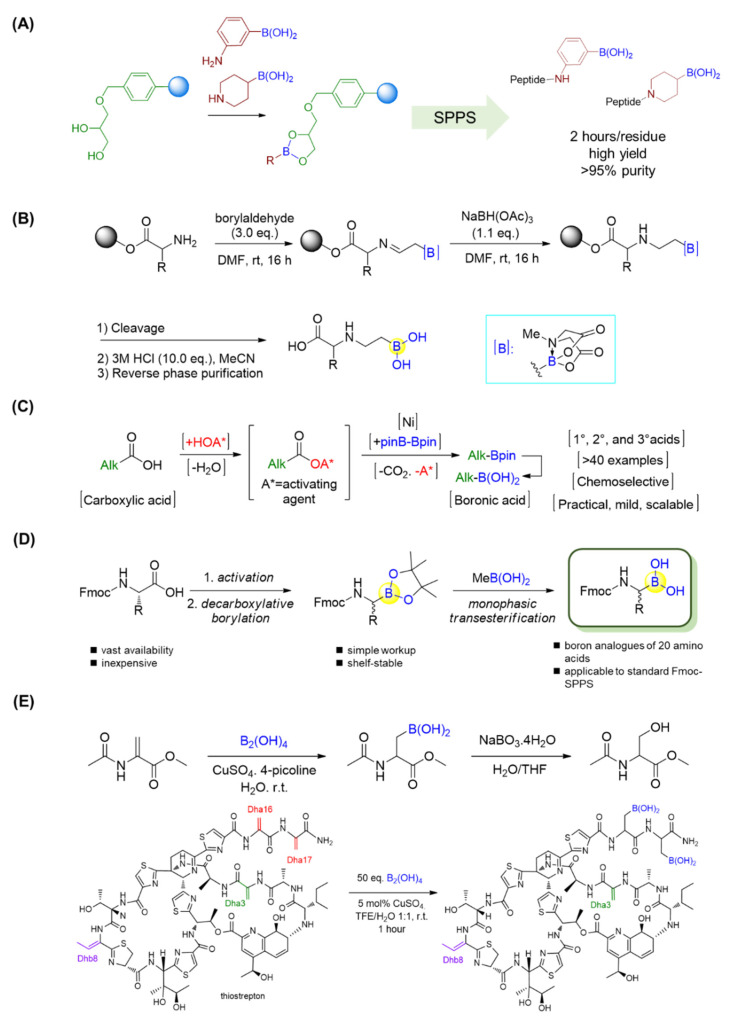
(**A**) Solid-phase synthesis of C-terminal boronic acid peptides; (**B**) Solid-phase synthesis of β-aminoboronic acid. (**C**) A nickel-catalyzed decarboxylative borylation enables the conversion of alkyl carboxylic acids into boronic acids; (**D**) Synthesis of Fmoc-α-aminoboronates for diversity-oriented peptide boronic acids synthesis; (**E**) β-borylation of dehydroalanine (Dha) residues.

## Data Availability

Not applicable.

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
