# Peer review of "Merging the Versatile Functionalities of Boronic Acid with Peptides"

_ijms, 2021, doi:10.3390/ijms222312958_

Round 1

Reviewer 1 Report

The Authors have written an interesting review able to give an overview of versatile use of boronic acid. The manuscript is well written and is clearly exposed. The examples chosen are enough to describe recent applications of boronic acid. 

I suggest to improve the pictures. A figure to show the inhibition mechanism of serine protease should be added. I suggest to divide the figure 2 in two images: the first one, compounds 1 to 6 should be converted in a table also indicating the pathology involved; the second image with compounds 7 to 9. In figure 3: add data of interaction between bortezomib and subunit b to underline the structural difference between peptides 10, 11, 12, 13 (if the data are available). In figure 4, the model of catalytic site of aspartyl protease should be added. In figure 5, I suggest to improve the figure legend enlarging the description. 

In figure 10, reaction D should be moved in 2nd position. The same change should be done in the text. In point E, there is a mistake in the reaction with maleimide.

Reviewer 2 Report

Dear authors,

Very nice and complete work around Boronic acid containing peptides and further applications of organoboron compounds on peptide synthesis/reactions etc. The present article can attract a wider readership interest and provide good knowledge around a wide spectrum of boron-peptide applications. I believe this work can be published in IJMS after minor changes, please mind the comment/suggestions bellow:

Comment/suggestion

  • Any reference to the empty 2p orbital of Boron?
  • Addition of a remark regarding bioisosterism of boronic acid group should be added. Take into consideration literature 10.1002/cmdc.201200585 and more recently 10.3390/molecules21091185.
  • I believe line 33 comment on peptide ease of synthesis is too general and not taking into consideration linearity, cyclic, sequence length etc…Hence it should be amended or further analyzed! Unless if it refers to a specific synthetic approach in which case it should be named accordingly (e.g. solid phase etc)…
  • Line 71, which 5 compounds? Structures should be displayed somewhere.
  • Line 127 is mentioned the term warhead which is used for covalent bond forming groups in medicinal chemistry….I believe it should be also mentioned in Introduction before anywhere else in the text for better clarity to the readership.
  • Line 157, provide again the figure representing Bortezomib chemical structure (i.e. Figure 1)
  • Line 165, the Figure 3 bracket should be moved elsewhere in order not to be confused as the place of where Bortezomib chemical structure is mentioned.
  • Line 215 should be rephrased “Arginases (two isoforms, ARG-1 and ARG-2)”…
  • Line 282 should be rephrased “These BPFSs may find great potential”…
  • Whole sentence lines 285-288 should be rephrased…

Best regards,
